# Excellent Performances of Composite Polymer Electrolytes with Porous Vinyl-Functionalized SiO_2_ Nanoparticles for Lithium Metal Batteries

**DOI:** 10.3390/polym13152468

**Published:** 2021-07-27

**Authors:** Hui Zhan, Mengjun Wu, Rui Wang, Shuohao Wu, Hao Li, Tian Tian, Haolin Tang

**Affiliations:** 1State Key Laboratory of Advanced Technology for Materials Synthesis and Processing, Wuhan University of Technology, Wuhan 430070, China; huiz@whut.edu.cn (H.Z.); wumengjun2019@whut.edu.cn (M.W.); 12779@whut.edu.cn (R.W.); wushuohao@whut.edu.cn (S.W.); leanne@whut.edu.cn (H.L.); 2Guangdong Hydrogen Energy Institute of WHUT Xianhu Hydrogen Valley, Foshan 528200, China

**Keywords:** lithium batteries, composite polymer electrolytes, chemical grafting, porous nanoparticles, cross-linking polymerization, silane coupling

## Abstract

Composite polymer electrolytes (CPEs) incorporate the advantages of solid polymer electrolytes (SPEs) and inorganic solid electrolytes (ISEs), which have shown huge potential in the application of safe lithium-metal batteries (LMBs). Effectively avoiding the agglomeration of inorganic fillers in the polymer matrix during the organic–inorganic mixing process is very important for the properties of the composite electrolyte. Herein, a partial cross-linked PEO-based CPE was prepared by porous vinyl-functionalized silicon (p-V-SiO_2_) nanoparticles as fillers and poly (ethylene glycol diacrylate) (PEGDA) as cross-linkers. By combining the mechanical rigidity of ceramic fillers and the flexibility of PEO, the as-made electrolyte membranes had excellent mechanical properties. The big special surface area and pore volume of nanoparticles inhibited PEO recrystallization and promoted the dissolution of lithium salt. Chemical bonding improved the interfacial compatibility between organic and inorganic materials and facilitated the homogenization of lithium-ion flow. As a result, the symmetric Li|CPE|Li cells could operate stably over 450 h without a short circuit. All solid Li|LiFePO_4_ batteries were constructed with this composite electrolyte and showed excellent rate and cycling performances. The first discharge-specific capacity of the assembled battery was 155.1 mA h g^−1^, and the capacity retention was 91% after operating for 300 cycles at 0.5 C. These results demonstrated that the chemical grafting of porous inorganic materials and cross-linking polymerization can greatly improve the properties of CPEs.

## 1. Introduction

Nowadays, the urgent demands for green energy and high-energy storage systems promote the high-speed development of rechargeable energy storage devices [1,2,3,4,5]. Lithium-ion batteries (LIBs) are regarded as one of the most hopeful candidates in the field of portable electronics, electric vehicles, and energy storage stations [6,7,8]. However, there are serious safety issues in traditional commercialized lithium-ion batteries, as the use of liquid electrolyte has the shortcomings of leakage, flammability, and toxicity [9]. Therefore, with the inevitable trend of upgrading lithium-ion batteries, solid-state electrolytes (SSEs) show huge potential in enhancing the safety performance of LIBs and have been researched extensively [10]. Moreover, SSEs make it possible to use the Li metal, which possesses high theoretical capacity (3680 mA h g^−1^) and the lowest electrochemical potential (−3.04 V vs. standard hydrogen electrode) as an anode electrode in lithium battery [11,12,13,14]. Lithium-metal batteries (LMBs) with SSEs have higher specific capacity and more stable electrochemical stability, and they can be used in wearable electronic devices [11,15,16].

Generally, there are three main categories according to the characteristic of SSEs: inorganic solid electrolytes (ISEs), solid polymer electrolytes (SPEs), and composite polymer electrolytes (CPEs) [17,18,19,20]. ISEs (e.g., SiO_2_, TiO_2_, LLZO, and LAGP) are well known by their mechanical rigidity and nonflammability. However, their fragile and hard properties could lead to severe solid–solid interface problems between the electrodes and electrolytes, which ultimately impedes their further application in LMBs [18,21]. On the contrary, SPEs have been studied and applied for their flexibility, processability, and excellent interface compatibility with electrodes. SPEs are usually prepared by dissolving lithium salts in polymers (e.g., PEO, PVDF, PAN, PMMA), in which Li^+^ are transported across polymer segments [22]. Unfortunately, they are suffering from low Li^+^ conductivity at room temperature and are far from commercialization [23,24,25]. CPEs combine the advantages of ISEs and SPEs, which are constructed by introducing ceramic materials into polymer matrixes to obtain remarkably ionic conductivity (≈10^−4^ S cm^−1^) to enhance the flexibility and good interfacial contact [26,27].

PEO-based CPEs have attracted a large amount of research since they were first applied in solid electrolytes by Armand et al. in 1978 [28]. PEO has excellent flexibility, an outstanding ability to dissolve lithium salts, and high ion conductivity at elevated temperatures [29,30]. However, pure PEO electrolyte with inferior mechanical stability is insufficient to restrain the formation of lithium dendrites and the intrinsic high crystallinity of PEO, which results in a poor ionic conductivity originally depending on the amorphous region [31,32]. Incorporating inorganic fillers into the PEO matrix is an effective way to gain flexible, nonflammable, and mechanically robust electrolytes [26,33]. The composite electrolyte acquired from a conventional mechanical mixing generally fails to manifest an enhanced ionic conductivity, mechanical property, and electrochemical performance. One crucial reason is that inorganic nanomaterials are easy to aggregate by increasing the content due to the strong specific surface energy. It is difficult to construct a well-percolated Li^+^-conductive network because of severe agglomeration and the low utilization of inorganic fillers [34,35].Therefore, some researchers equipped inorganic nanoparticles with functional groups to combine fillers and polymer matrixes by chemical crossing and indicated significant potential in this area. On the basis of Lewis acid–base model interaction, chemical bonding is more favorable for increasing the ionic conductivity of CPEs because of (1) preventing PEO reorganizing and increasing the proportion of the amorphous area; (2) helping further dissociating of Li salt and plasticizing the system through the Lewis acid center on fillers; (3) providing more ion transport pathways in the surface and inside of inorganic fillers as well as interfaces between the fillers and PEO chains [36,37]. Nan’s group prepared a PVDF-based CPE utilizing dehydrofluorination catalyzed by La of LLZTO which was only suitable for small amounts of polymers, such as PVDF and PVDF-HFP [38]. Yang’s group proposed a strategy for synthesizing nano-sized SiO_2_ particles in situ within PEO matrix through acid–base interaction and hydrogen bonding, showing an improved ionic conductivity (≈1.1 × 10^−4^ S cm^−1^ at 30 °C) [39]. Furthermore, functionalized mesoporous silicon materials have been widely researched in the past decades due to the large specific surface area and controllable microstructure, which endow them with great potential for applications in CPEs [40]. Kim’s group introduced mesoporous organosilica into PEO and the CPE demonstrated an ion transfer number up to 0.9 [41]. However, few people had combined chemical bonding and functionalized mesoporous materials in a CPE. Inspired by these previous research studies, herein, we developed a partial cross-linked PEO-based composite solid electrolyte with porous vinyl-functionalized (p-V-SiO_2_) silicon nanoparticles as fillers and poly (ethyleneglycol diacrylate)(PEGDA) as cross-linkers. The synthesis of p-V-SiO_2_ nanoparticles was based on a simple one-step method from Stein [42,43]. In this system, the chemical bonding of inorganic/organic materials as well as the special morphology and structure of fillers would play a synergistic role to improve the performance of the CPEs. SiO_2_ nanoparticles could compensate for the poor mechanical properties of PEO and inhibit PEO recrystallization, while its large specific surface area and a mass of pores could provide sites for highly reactive vinyl and allow the permeation of organic chains. Except as cross-linkers, PEGDA was compatible to PEO and could act as a plasticizer in the system. As a result, the as-made flexible CPE exhibited higher ionic conductivity, a wider electrochemical stability window, and better ability to suppress dendrite growth. Ultimately, the solid-state Li|CPE|LiFePO_4_ cells demonstrated excellent cycling performance and electrochemical stability.

## 2. Materials and Methods

### 2.1. Materials

Tetraethyl orthosilicate (TEOS), sodium hydroxide (NaOH), anhydrous acetonitrile (ACN), anhydrous ethanol, and aqueous (HCl) (37%) were purchased from Sinopharm Chemical Reagent (China). Cetyltrimethylammonium bromide (CTAB), poly (ethylene glycol) diacrylate (PEGDA, *M*w = 400), polyethylene oxide (PEO, *M*w = 1,000,000), polyvinylidene fluoride (PVDF), Super P and N-Methyl pyrrolidone (NMP) were purchased from Aladdin Reagent (Shanghai, China). Bis (trifluoromethyl) sulfonamide lithium salt (LiTFSI) (99.95%), LiFePO_4_, and vinyltriethoxysilane (VTES) were purchased from Sigma-Aldrich (Tianjin, China).

### 2.2. Synthesis of Porous Silica Nanoparticles (p-SiO_2_)

P-SiO_2_ was synthesized by the hydrolytic condensation of TEOS. First, 0.5 g (1.37 mmol) of CTAB and 0.08 g (2 mmol) of NaOH were dissolved in a mixture solution of deionized water (240 mL) and ethanol (40 mL). The mixture was ultrasonic dispersed for 10 min. Then, 3 mL (13.4 mmol) of TEOS was dropped into the reaction mixture at a rate of 0.1 mL/min under stirring at 80 °C for 3 h. Then, the product was centrifuged, washed, and freeze-dried. The CTAB was extracted by stirring in acid/solvent extraction, using a solution of 50 mL of ethanol and 0.5 mL of aqueous HCl (37%) per 0.5 g of sample for 24 h at 80 °C. The product was centrifuged, washed, and freeze-dried for 24 h.

### 2.3. Synthesis of Porous Vinyl-Functionalized Silica Nanoparticles (p-V-SiO_2_)

In order to synthesize p-V-SiO2, 0.6 g (1.65 mmol) of CTAB and 0.1 g (2.5 mmol) of NaOH were dissolved in a mixture of deionized water (250 mL) and ethanol (50 mL). The mixture was ultrasonic dispersed. Then, a mixed solution of TEOS (3 mL/0.013 mol) and VTES (0.75 mL/0.0036 mol) was dropped into the reaction mixture at a rate of 0.1 mL/min under stirring. The resulting mixture was stirred for 1 h at room temperature, which was followed by heating at 80 °C for 24 h under static state. The product separation and template removal are consistent with the methods mentioned above.

### 2.4. Preparation of Composite Solid Electrolyte Membranes (CSEs)

Typically, 0.5 g of lithium LiTFSI, p-SiO_2_(or p-V-SiO_2_) powder with different weight ratios (5%, 8%, 10% or 15% based on the total mass of PEO and PEGDA), 0.5%(based on the total weight of monomers) initiators (AIBN), and 0.3 g of monomers (PEGDA, *M*w = 400) were mixed and dissolved in ACN, which was followed with sonication for 10 min to obtain uniform dispersion. Then, 1.2 g of poly (ethylene oxide) (PEO, *M*w = 1,000,000) was added into the solution and then stirred at room temperature for 24 h to get homogeneous, translucent, and viscous precursor solution. The resultant solution was cast onto a flat Teflon mold with a scraper and then dried for 12 h at room temperature and polymerized for 24 h at 60 °C in a vacuum-drying oven. The PEO/LiTFSI SPE membranes were fabricated in the similar way. The thickness of electrolyte film was controlled at about 120 μm. All the solid electrolytes were stored in a glove box filled with argon gas for more than 24 h before further testing.

### 2.5. Cathode Preparation and Cell Construction

The battery electrochemical performance was evaluated using CR2016 coin cells. For fabricating Li|CPEs|LiFePO_4_ cells, LiFePO_4_, PVDF, and Super P were hybrided in NMP solution with a mass ratio of 7:1.5:1.5. Then, the resultant slurry was cast on an Al foil current collector and dried at 80 °C for 12 h under vacuum to remove the residue solvent. The loading of LFP was about 1.0 mg cm^−2^. Li metal served as the anode, and the as-prepared CSEs membranes served as the electrolyte and separator. All cell fabrications were operated in a glove box filled with argon gas.

### 2.6. Material Characterization

The X-ray diffraction (XRD) was performed on a D/MAX-RB diffractometer (Rigaku, Tokyo, Japan) using a Cu Kα radiation (λ = 0.1541 nm) with 2θ from 0° to 10°. N_2_ adsorption measurements were performed with an ASAP2020 adsorption analyzer (Micromeritics, Atlanta, Georgia, USA) to measure the surface areas and pore sizes. The top and cross-section scopes of the CPE membranes were characterized by scanning electron microscopy (SEM, Hitachi JEM-7500F, ZEISS, Jena, Germany) and transmission electron microscopy (TEM, JEM-2100F, JEOL, Tokyo, Japan); the elemental distribution was analyzed by energy-dispersive X-ray spectroscopy (EDS, ZEISS, Jena, Germany). The chemical structure of the samples was demonstrated through Fourier transform infrared spectroscopy (FT-IR) on a Nicolet6700 spectrometer (Therno Nicolet, Madison, Wisconsin, USA). TGA analysis (NETZSCH-STA449F3 analyzer, Selb, Germany) of the samples was carried out from 30 to 800 °C under N_2_. The mechanical properties of the as-prepared membranes were obtained from tensile tests using a stretch testing machine (CMT6202, SAAS, Shenzhen, China) at a stretching speed of 50 mm·min^−1^.

### 2.7. Electrochemical Measurements

The ion conductivities (σ) of the as-prepared electrolytes were calculated by electrochemical impedance spectroscopy (EIS) with the frequency ranging from 1 MHz to 0.01 Hz on a CHI660E electrochemical station (CHInstruments, ShangHai, China). Stainless steels (SS) were used as blocking electrodes to construct a SS|CPEs|SS sandwiched structure. The calculation formula is
(1)$$\sigma =\frac{L}{\;\mathrm{RS}\;}$$
where L, R, and S represent the thickness of the membranes, the area of the SS plates, and the bulk resistance, respectively. Liner sweep voltammetry (LSV) was recorded to test the electrochemical stability of CSEs in the Li|SS cells with the CHI660E electrochemical station at 0.5 mV·s^−1^ at 60 °C in a voltage ranging from 0 to 7 V. Galvanostatic cycling of symmetric Li|CPEs|Li cell was detected at 60 °C to evaluate the compatibility between CPEs and lithium metal electrode, and the ability to inhibit lithium dendrites information of the CPEs on a Land battery test system. The electrochemical performance of Li|CPEs|LiFePO_4_ cells were analyzed on a Land battery test system in a potential range of 2.5 to 4.2 V at 60 °C to evaluate the cycling and rate capabilities. Figure 1a illustrates the entire synthetic route of p-V-SiO_2_/PEO cross-linked composite polymer electrolyte. First, p-V-SiO_2_ was synthesized by a hydrolytic condensation of TEOS and VTES in a basic solution with CTAB serving as surfactant. After removing the CTAB, an ordered mesoporous structure has been formed in the SiO_2_ nanoparticles where they were covered with vinyl groups. P-SiO_2_ nanoparticles were produced by the similar way without using the silane coupling agent. After that, the produced p-V-SiO_2_ (or p-SiO_2_) and PEGDA were cross-linking through thermal addition polymerization in a PEO host with a certain amount of LiTFSI (EO/Li^+^ = 20) to form inorganic/organic CPEs. Figure 1b showed the prepared white p-V-SiO_2_ powder. Figure 1c demonstrated the 10% p-V-SiO_2_/PEO CPE precursor solution, which was opalescent and viscous.

## 3. Results

The small-angle XRD patterns of p-V-SiO_2_ and p-SiO_2_ are shown in Figure 2a. The observation of one sharp peak and two weak peaks from p-SiO_2_ indicate the presence of d_100_, d_110_, and d_200_ reflections, which could be indexed to the hexagonally ordered pore structure of MCM-41 [42,43,44]. Although the absorption peaks shifted to the left along with a decrease of the intensity, the p-V-SiO_2_ still showed similar structural characteristics, which may be ascribed to the damage of the integrity by grafted vinyl. The nitrogen adsorption isotherms (Figure 2b) and pore diameter distribution of the two kinds of SiO_2_ samples demonstrated a representative type IV isotherm for obvious mesoporous structure. Corresponding to the results of XRD, the p-V-SiO_2_ had a narrower hysteresis loop, smaller specific surface area, and pore volume than p-SiO_2_ due to grafted vinyl similarly. The insert image also demonstrated a narrow pore diameter distribution of p-SiO_2_ which might be attribute to the fact that vinyl groups occupied part of the channel. The attachment of vinyl groups within the channels had been proved by a bromination reaction in some previous studies [43,45,46]. The results of the specific surface areas were determined by the BET method, pore volumes were gained from a single point of adsorption, and pore sizes were obtained from the BJH model, as summarized in Table 1 [42,43].

The TGA data and FT-IR spectra (see Appendix A) could confirm the complete removal of CTAB, successful grafting of vinyl, and full polymerization of monomers. The different weight loss ratios of p-V-SiO_2_ and p-SiO_2_ in TGA thermograms indicated that vinyl groups were decorated on nano-SiO_2_ and started to decompose at about 250 °C. This was similar to the phenomenon reported in the previous literature [43,44]. In the FT-IR spectra, the absorption peaks at 3467 cm^−1^, 2900 cm^−1^, 1750 cm^−1^, and 1215 cm^−1^ represented the associating O-H of SiO_2_ nanoparticles as well as stretching vibration of C–H, C=O, and C-O bonds, respectively. Moreover, the peak at 1660 cm^−1^ was related to the C=C, and 1120 cm^−1^ and 880 cm^−1^ were assigned to the symmetrical and asymmetrical stretching vibration of Si–O–Si [25,34,42,45].

As shown in Figure 3, the morphologies, sizes, and elemental distribution of p-V-SiO_2_ (Figure 3a,b,e) and p-SiO_2_ (Figure 3c,d) were investigated via TEM and EDS. These images verified the obvious porous morphology and spherical shape with average diameter of about 200 nm. The grafted SiO_2_ nanoparticles were slightly deformed and had an inhomogeneous distribution in size. Energy-filtered TEM (EFTEM) further revealed the mesoporous structure, particle size, and average distribution of Si, O, and C (from C=C) elements.

As shown in Figure 4, the top and cross-section views of the CPE membranes from SEM showed a smooth and homogenous surface of the CPEs (Figure 4a,d). In the magnified images (Figure 4b,e), more wrinkles and cracks appeared in the 10% p-SiO_2_/PEO CPE, which proved that functionalization with the vinyl and mesoporous structure of SiO_2_ indeed help eliminate aggregation, promote the dissociation of Li salt, and form a tight inorganic/polymer composite network. The thickness of the films was about 120 μm, and no significant phase separation or aggregation was observed in the cross-section (Figure 4c,f). The membranes were white and transparent, with excellent flexibility for being bent and twisting randomly (Figure 4g–i). The excellent flexibility of the CPE membranes could also be proved by Figure 5a, which depicted ultrahigh 650% maximum elongation. This value was much higher than that reported in other studies in the literature [25,27,34,39].

The safety and availability of LMBs operating under long-term and extreme conditions are largely dependent on the thermal stability and mechanical strength of the electrolytes. The thermal stability of the CPEs was examined by TGA and depicted in Appendix A; p-SiO_2_/PEO CPE encountered a sharp decomposition around 220 °C, unlike the 10% p-V-SiO_2_/PEO CPE, which did not start to degrade until 300 °C. This might be due to the large surface area of p-V-SiO_2_ and stable composite cross-linked network, which could reduce thermal resistance and favor heat dissipation. The fact was also been demonstrated before [39,47].

The stress-strain curves of CPEs with different mass ratios of p-V-SiO_2_ (or p-SiO_2_) are depicted in Figure 5a. The 15% p-V-SiO_2_/PEO CPE manifested the largest tensile strength up to 2.46 MPa compared to 2.27 MPa of 10% p-V-SiO_2_/PEO CPE and 2.24 MPa of 12% p-V-SiO_2_/PEO CPE. However, the latter two CPEs possessed better flexibility with 650% and 830% maximum elongation, respectively. Nevertheless, the 10% p-SiO_2_/PEO CPE only performed a tensile strength of 1.84 MPa and maximum elongation of 230%. This phenomenon might stem from the chemical bonding of organic and inorganic materials, which was favorable to this combination and avoided the agglomeration of particles. The 10% p-V-SiO_2_/PEO CPEs operated at 60 °C were selected as the most ideal samples, and the working conditions after the ionic conductivity of all CPEs were systemically studied by EIS in the temperature range of 30 to 80 °C. The results of temperature-dependent ionic conductivity and the consistent Arrhenius plots were plotted in Figure 5b and Appendix A. Over the entire temperature range, 10% p-V-SiO_2_/PEO CPE exhibited the highest ionic conductivity (5.08× 10^−4^ S cm^−1^ at 60 °C), while the 10% p-SiO_2_/PEO CPE had a smaller one (1.3× 10^−4^ S cm^−1^ at 60 °C). The high ionic conductivity originated from the presence of vinyl grafting layers on porous SiO_2_, which enabled cross-linking polymerization between fillers and PEGDA in the PEO host. Therefore, the severe agglomeration of inorganic materials could be avoided. Lithium salt was dissolved further and PEO recrystallization was inhibited due to the large specific surface area as well as the large number of pores of SiO_2_ nanoparticles [34]. Another crucial parameter to determine the possible application in LMBs is the electrochemical stability window. LSV measurements were carried out at 60 °C and are illustrated in Figure 5c. The 10% p-V-SiO_2_/PEO CPE started oxidative decomposition at 5.2 V and 5 V for 10% p-SiO_2_/PEO CPE. The improved electrochemical stability of the former might be attributed to the higher ionic conductivity to reduce interfacial overpotential and better compatibility between ceramic and polymers because of chemical bonding. Effectively inhibiting the dendrite growth of lithium was the most important capability of solid electrolyte used in LMBs. Li plating (0.5 h)/stripping (0.5 h) experiments with Li metal symmetric electrodes were made at 60 °C, and the time-voltage profiles are shown in Figure 5d,e. Both the 10% p-V-SiO_2_/PEO CPE and 10% p-SiO_2_/PEO CPE could stably work in cycles under the relatively low current density of 0.1 mA cm^−2^ and 0.2 mA cm^−2^. When the current density was improved to 0.3 mA cm^−2^, the 10% p-V-SiO_2_/PEO CPE could remain stable over 450 h with the overpotential of about 600 mV, whereas an obvious increased polarization was observed in 10% p-V-SiO_2_/PEO CPE following a short circuit after 250 h. These results suggested that compared to others, 10% p-V-SiO_2_/PEO CPE had a stronger ability to inhibit the lithium dendrite information and to homogenize lithium deposition with the help of rigid porous functional inorganic fillers. Benefiting from the cross-linked composite network, the lithium ion flow was more uniform, and the interface compatibility was improved [27,34,35].

These advantages of p-V-SiO_2_/PEO CPE also contributed to the performances of Li|CPEs|LiFePO_4_ cells. The rate capability at different current densities (0.1–2 C) were assessed at 60 °C. Figure 6a demonstrated that the discharge-specific capacities of Li|p-V-SiO_2_/PEO CPE|LiFePO_4_ cells were 169.1, 158.6, 153.4, 135, and 109.9 mA h g^−1^ at 0.1, 0.2, 0.5, 1, and 2 C, respectively. Even after experiencing a current density of 2 C, the specific capacity could nearly recover to the initial state of 165.5 mA h g^−1^ when it returned to 0.1 C. However, the specific capacities of Li|10% p-SiO_2_/PEO CPE|LiFePO_4_ decayed rapidly with the increase of current density and suffered a short circuit at 1 C (so, the specific capacities at 1 C and 2 C were zero). The specific capacity could not recover to the initial value. Figure 6b showed a smooth discharge platform and a slow increasing polarization of Li|10% p-V-SiO_2_/PEO CPE|LiFePO_4_ battery, indicating that no other side reactions occurred during the charge and discharge processes. Figure 6c further proves the excellent cyclability of Li|LiFePO_4_ cells with p-V-SiO_2_/PEO CPE at 0.5 C under 60 °C. Its first discharge-specific capacity was 155.1 mA h g^−1^, and the coulombic efficiency was above 99.5%. Moreover, there was only a capacity degradation of 9.1% after 300 cycles. The insert image showed the single cycle charge–discharge curves of Li|10% p-V-SiO_2_/PEO CPE|LiFePO_4_, and Appendix A showed that it could cycle stably at 1 C with a capacity retention of 69% after 150 cycles. In addition, Figure 6d,e demonstrate the SEM images of Li electrodes from Li|LiFePO_4_ cells with 10% p-V-SiO_2_/PEO CPE and 10% p-SiO_2_/PEO CPE after 150 cycles, respectively. Many dendrites appeared on the tough surface of Li piece in Li|10% p-SiO_2_/PEO CPE|LiFePO_4_, while the surface of the Li anode adapted with 10% p-V-SiO_2_/PEO CPE was smooth. It could be summarized that the P-V-SiO_2_-based CPEs possessed an excellent capacity to impede Li dendrite growth, boost effective transportation lithium ion, and establish a more compatible interface. All of the above results revealed the superior reversibility, outstanding stability, and high discharge-specific capacity of the 10% p-V-SiO_2_/PEO electrolytes [10,34,35]. 

## 4. Conclusions

In conclusion, PEO-based composite polymer electrolytes with porous vinyl-functionalized silicon (p-V-SiO_2_) nanoparticles were prepared (the 10% p-V-SiO_2_/PEO CPE was proved to the most ideal sample) which enabled (1) the cross-linking of inorganic filler and cross-linking agent (PEGDA) in the PEO host, (2) a configuration of an interconnected network that could successfully eliminate the agglomeration effects typically occurred in mechanical mixing, (3) uniform distribution of porous SiO_2_ nanoparticles and further dissolution of lithium salt benefiting from the formation of large specific area and chemical bonding, (4) great contributions to the high mechanical robustness and flexibility (2.27 MPa tensile strength and 650% maximum elongation), (5) improved ionic conductivity (5.08 × 10^−4^ S cm^−1^ at 60 °C), (6) superior ability of suppressing lithium dendrite growth of the CPEs, (7) a wide electrochemical window (up to 5.2 V at 60 °C). Moreover, the fabricated Li|CPE|LiFePO_4_ exhibited an excellent rate property (169 mA h g^−1^ at 0.1 C and 110 mA h g^−1^ at 2 C) and could recover to the initial state quickly even after 2 C. It also demonstrated outstanding cycling performances for 300 cycles with a capacity retention of 91% at 0.5 C, and the first discharge specific capacity was 155.1 mA h g^−1^. To sum up, this strategy about chemical combining inorganic/organic materials as well as utilizing the special morphology and structure of fillers has huge potential in the development of improved functional CPEs membranes. Of course, there are some shortcomings in the CPEs, the most important of which is to further improve the ionic conductivity of the CPEs at room temperature in order to reduce the operational temperature of the electrolyte film. In future research, some good methods need to be proposed.

## Figures and Tables

**Figure 1 polymers-13-02468-f001:**
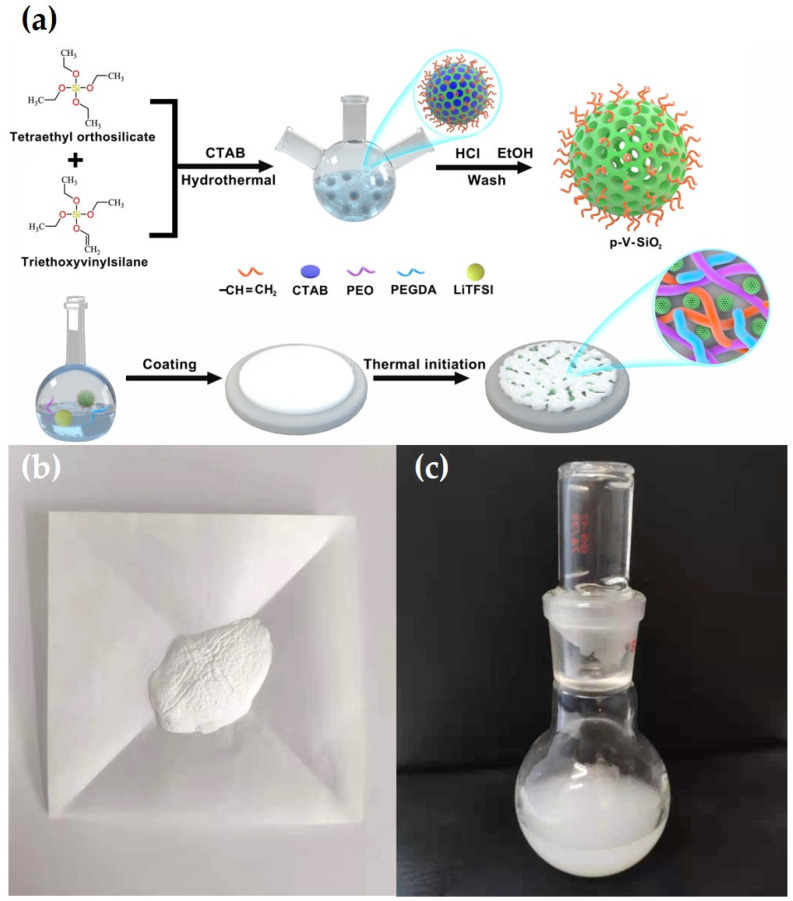
(**a**) Schematic illustration for the synthetic route of p-V-SiO_2_/PEO cross-linked CPE. (**b**) The photograph of p-V-SiO_2_ powder. (**c**) The photograph of 10% p-V-SiO_2_/PEO CPE precursor solution.

**Figure 2 polymers-13-02468-f002:**
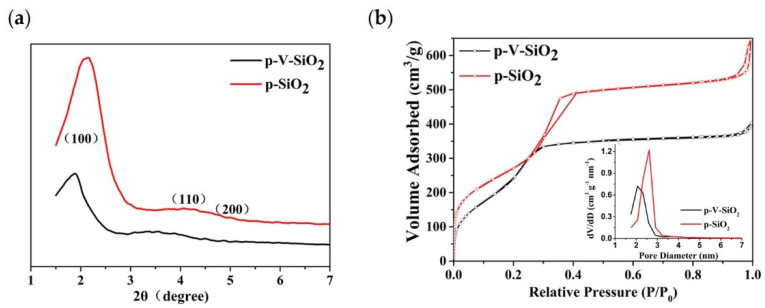
(**a**) Small-angle XRD patterns of p-V-SiO_2_ and p-SiO_2_. (**b**) Nitrogen adsorption isotherms of p-V-SiO_2_ and p-SiO_2_. The inset demonstrates the pore diameter distribution of p-V-SiO_2_ and p-SiO_2_.

**Figure 3 polymers-13-02468-f003:**
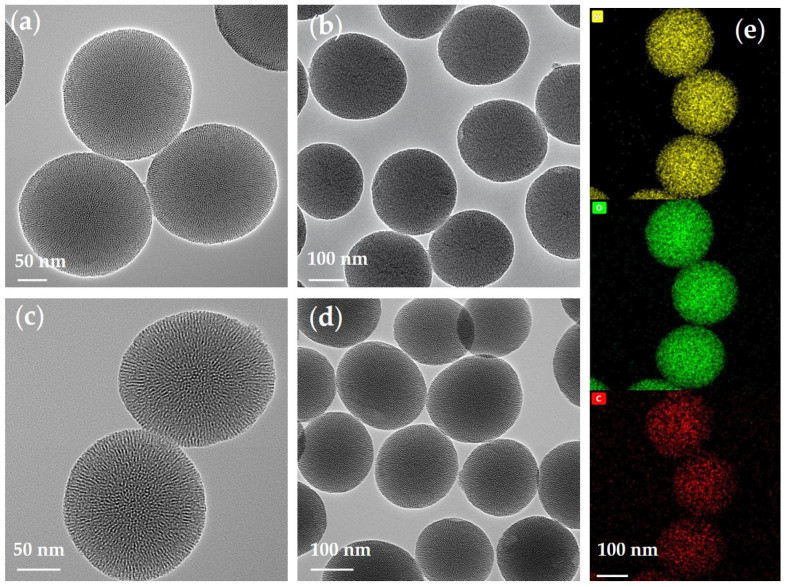
(**a**,**b**) TEM images of p-V-SiO_2_. (**c**,**d**) TEM images of p-SiO_2_. (**e**) Si, O, and C elements in p-V-SiO_2_.

**Figure 4 polymers-13-02468-f004:**
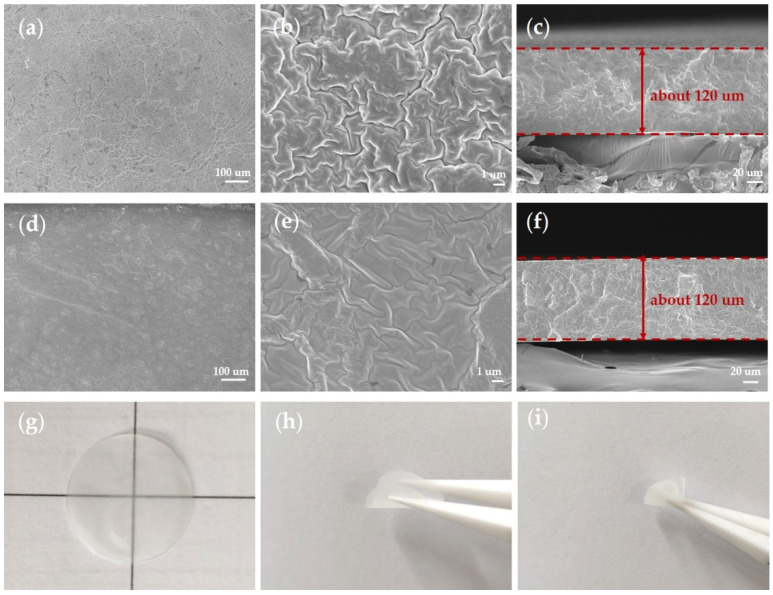
(**a**–**c**) SEM images of 10% p-V-SiO_2_/PEO CPE. (**d**–**f**) SEM images of 10% p-SiO_2_/PEO CPE. (**g**) Optical images of the 10% p-V-SiO_2_/PEO CPE. (**h**) bent 10% p-V-SiO_2_/PEO CPE. (**i**) twisted 10% p-V-SiO_2_/PEO CPE.

**Figure 5 polymers-13-02468-f005:**
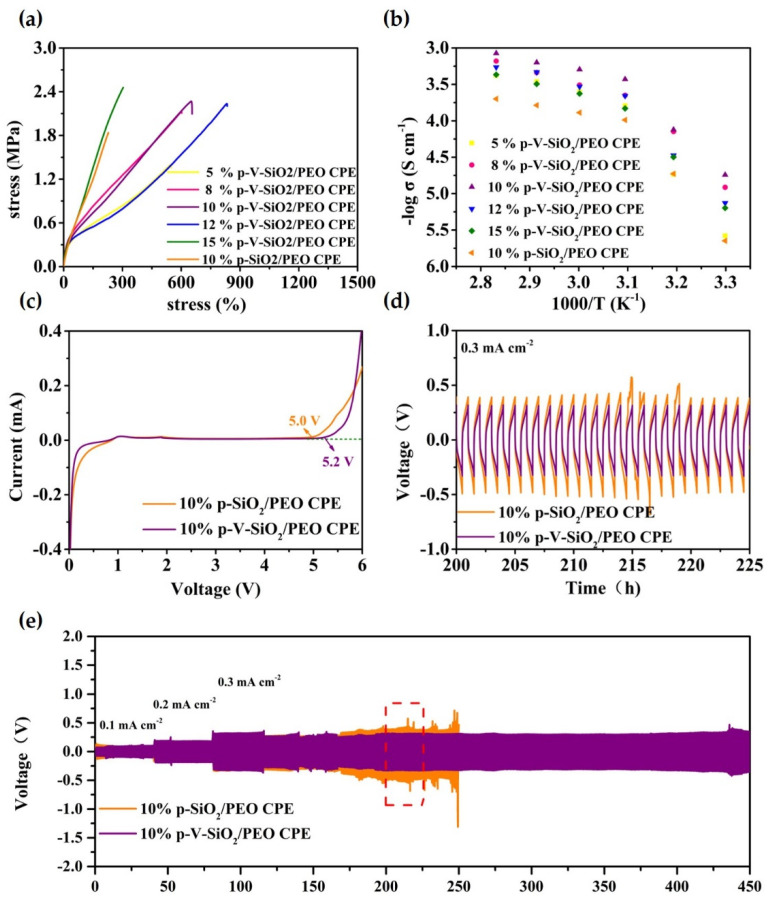
(**a**) Stress−strain curves of CPEs with different mass ratios of p-V-SiO_2_ (or p-SiO_2_). (**b**) EIS profiles of CPEs with different mass ratios of p-V-SiO_2_ (or p-SiO_2_) at the range of 30−80 °C. (**c**) LSV curves of 10% p-V-SiO_2_/PEO CPE and 10% p-V-SiO_2_/PEO CPE membranes at 60 °C. (**d**) The enlarged time–voltage profiles from 200th to 225th h. (**e**) The time–voltage profiles of Li|10% p-V-SiO_2_/PEO CPE|Li and Li|10% p-SiO_2_/PEO CPE|Li symmetrical cells at 60 °C.

**Figure 6 polymers-13-02468-f006:**
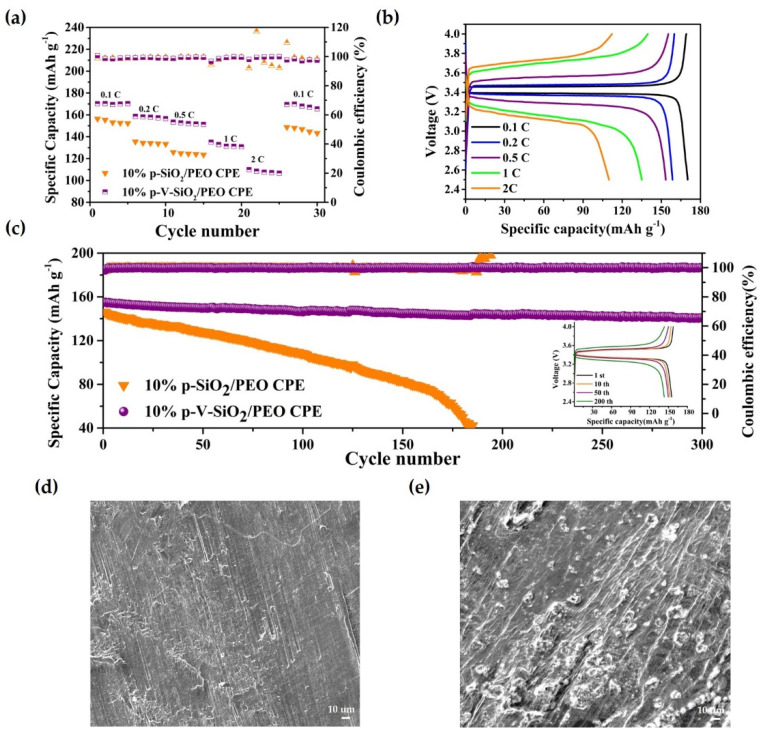
(**a**) Rate capability at different current densities at 60 °C. (**b**) Charge–discharge profiles of Li|10% p-V-SiO_2_/PEO CPE|LiFP cells at different current densities at 60 °C. (**c**) Cycle performance of Li|10% p-V-SiO_2_/PEO CPE|LiFP cells at 0.5 C at 60 °C. The insert image is the charge–discharge profiles of Li|10% p-V-SiO_2_/PEO CPE|LiFP cells under different cycle numbers. (**d**,**e**) The SEM images of Li electrodes from Li|LiFePO_4_ cells with 10% p-V-SiO_2_/PEO CPE and 10% p-SiO_2_/PEO CPE after 150 cycles, respectively.

**Table 1 polymers-13-02468-t001:** The data of specific surface areas, pore volumes, and pore sizes of p-V-SiO_2_ and p-SiO_2_.

Sample	BET Surface Area (m^2^/g)	Pore Volume (cm^3^/g)	Pore Diameter (nm)
p-SiO_2_	999	0.94	3.1
p-V-SiO_2_	947	0.61	2.4

## Data Availability

The data presented in this study are available in this study and the corresponding Appendix A.

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
