# Peer review of "Excellent Performances of Composite Polymer Electrolytes with Porous Vinyl-Functionalized SiO2 Nanoparticles for Lithium Metal Batteries"

_polymers, 2021, doi:10.3390/polym13152468_

Round 1

Reviewer 1 Report

  1. The manuscript is well organized, and the subject is well presented. The methods used are sound and the presentation and discussion of results is logical.
    The manuscript requires some major revisions to bring it to a level worthy of publication. My recommendations are detailed below:
  2. The current study investigates the use of composite polymer electrolytes combined with Vinyl-functionalized SiO2 nanoparticles for use in batteries. For this, the authors aim to avoid the agglomeration. The authors report that the cell can operate up to 450 hours without problems and the developed batteries showed outperforming cycling performance with a capacity retention of 91% after 300 cycles.
  3. The abstract is clear, but please consider reviewing the abstract and highlight the novelty, major findings and conclusions. Perhaps line 17-18 can be removed (suggested).
  4. The literature review is limited in the introduction section, please report on past studies similar to this work or closely related to it, mention what they did and what were their main findings. Then explain how your current work brings new knowledge and difference to the field.
  5. In the last paragraph, the authors should consider answering the following question: what is the research gap did you find from the previous researchers in your field? Mention it properly. It will improve the strength of the article.
  6. The materials and methods section lacks any images of the experimental setup, equipment used or the fabricated samples! This is an experimental study and detail images are required to give a better idea about the project.
  7. Line 159-169 please combine all small paragraphs into larger one, check this issue in the rest of the article.
  8. Line 171-178 and scheme 1 these seem to be more suitable as part of the materials and method section?
  9. Please use only Figure and Table captions do not scheme unless this is allowed by the journal article format.
  10. Line 192 attributed
  11. Line 192 “attribute to the fact that vinyl groups occupied part of the channel” this is a speculation, the authors should compare this claim with results from the open literature similar to this work and verify whether they agree/disagree with this claim.
  12. Line 204-205 this sentence needs to be supported by a reference otherwise it is just a speculation. If this is a common fact, then it should be referenced properly.
  13. Line 225 why the authors use the letter u as a symbol for micron instead of the Greek symbol µ? This is not acceptable, please make sure to use proper symbols where necessary.
  14. Line 227-228 this sentence needs to be rephrased; also, how do you define excellent? What is the standards or threshold to say this membrane have excellent flexibility or average or poor? It is better to quantify such metrics rather than using words such as excellent.
  15. Line 237-239 this claim needs to be supported with references from studies in the open literature.
  16. Enlarge figure 4.
  17. Perhaps consider writing bullet points in the conclusion instead of a paragraph, 1-2 bullet points from each of the subsections in the results and discussion part.
  18. Some of the results are merely described and is limited to comparing the experimental observation. The authors are encouraged to include more discussion and critically discuss the observations from this investigation with existing literature.

Reviewer 2 Report

Summary:

The Research Article polymers 1314611 titled, “Excellent Performances of Composite Polymer Electrolytes with Porous Vinyl-functionalized SiO2 Nanoparticles for Lithium Metal Batteries,” reports a composite polymer electrolyte consisting of solid polymer electrolytes and inorganic solid electrolytes. The partial cross-linked PEO-based CPE as an electrolyte membrane has a high mechanical robustness and flexibility as well as an ionic conductivity of 5.08 x 10-4 S cm-1.

General comment:

In general, this research reports an interesting composite polymer electrolyte. A symmetric Li|CPE|Li cell and an all solid Li|LiFePO4 batteries are made to complete the electrochemical analysis. A minor revision is therefore suggested.

Comments:

(1) In the manuscript, the full names and the abbreviations in the parentheses should keep a space. The submission has some places with a space and some place without.

(2) What is the mass of the composite polymer electrolyte? It is suggested to consider the ratio of the composite polymer electrolyte and the loading of LFP.

(3) In the all solid Li|LiFePO4 batteries, what would happen if the loading of LFP increases?

(4) From the SEM images, the ordered mesopores seem to show preferred orientation. It is an interesting microstructure. Could the authors give more professional comments on this phenomenon if it would affect the electrochemical and battery performance.

(5) It is suggested to summarize the recent work in the development of composite polymer electrolyte in lithium-ion batteries and solid lithium-ion batteries.
Advanced Functional Materials 31 (2021) 2101380
Polymers 13 (2021) 535
Journal of Power Sources 482 (2021) 228929

Round 2

Reviewer 1 Report

All questions answered